# Identity information based on human magnetocardiography signals

## Abstract

We have developed an individual identification system based on magnetocardiography (MCG) signals captured using optically pumped magnetometers (OPMs). Our system utilizes pattern recognition to analyze the signals obtained at different positions on the body, by scanning the matrices composed of MCG signals with a $2 \times 2$ window. In order to make use of the spatial information of MCG signals, we transform the signals from adjacent small areas into four channels of a dataset. We further transform the data into time-frequency matrices using wavelet transforms and employ a convolutional neural network (CNN) for classification. As a result, our system achieves an accuracy rate of 97.04% in identifying individuals. This finding indicates that the MCG signal holds potential for use in individual identification systems, offering a valuable tool for personalized healthcare management.

## Introduction

In contemporary society, identification (ID) systems have become critical components in maintaining social order and ensuring security. The effectiveness of such systems is largely dependent on their ability to accurately and reliably identify individuals based on a secure and stable physiological feature. While several features, including facial features, fingerprints, iris patterns, and electrocardiogram (ECG) signals, have been employed for identification purposes (Batool and Tariq 2011; Su et al. 2019), each of these methods presents its own set of limitations and potential security concerns. For instance, facial features and fingerprints can be altered or obscured through various means, such as plastic surgery or wearing gloves. Similarly, iris patterns, although unique to each individual, can be challenging to capture accurately and reliably in certain situations, such as in low light or with individuals who have visual impairments. Moreover, ECG signals can be impacted by factors such as physical activity, medication, or stress, leading to variability and potential inaccuracies in identification. The identification of secure and stable physiological features for use in ID systems remains an active area of research, as the limitations of current methods continue to pose challenges for ensuring the accuracy and reliability of these systems.

Recent studies have demonstrated that the electric and magnetic signals generated by cardiac activity exhibit unique characteristics that hold promise for individual iden-

tification (Kim, Kim, and Pan 2020). Among these signals, the electrocardiogram (ECG) has been extensively investigated for its high security and potential use in identification systems (Biel et al. 2001; Israel et al. 2005). Typically, an ECG-based identification system involves two main components: pre-processing and deep learning (ElRahman 2019). However, one of the major challenges in using ECG signals for identification is the presence of noise, including industrial frequency noise, baseline wander, and Gaussian noise. Consequently, effective filtering techniques are essential to remove noise from ECG signals.

In some studies, ECG signals are transformed into time-frequency matrices using the wavelet transform or short-time Fourier transform to facilitate their analysis and classification (Byeon, Pan, and Kwak 2020). Neural networks, particularly convolutional neural networks (CNNs), are then trained on the 2-D time-frequency maps to identify individuals based on their unique ECG signals. Despite these promising developments, the practical application of ECG-based identification systems remains limited by the difficulty in collecting reliable ECG signals. This challenge is due in part to the fact that ECG signals are sensitive to various factors, such as physical activity, medication, and emotional states, which can impact the accuracy and consistency of the signals. Therefore, continued research and development efforts are needed to address these limitations and further enhance the feasibility and effectiveness of ECG-based identification systems

Magnetocardiography (MCG) is a noninvasive technique measuring the magnetic flux densities produced by electrical currents generated by the cardiomyocytes in the heart and the possible induced medium. MCG has been applied in medical diagnosis (Kwon et al. 2010), and further research is required to use MCG signals for identification purposes. MCG signals, which can be measured without contact or even through clothing (Tolstrup et al. 2006), present a potential physiological feature for an ID system. MCG has several advantages, including higher fraud resistance, weaker environmental dependence, and contactless measurement (Fenici and Melillo 1993), making it a promising candidate for individual identification. We collect MCG signals with our self-manufactured OPMs. Our OPMs are based on an original Bell-Bloom configuration (Sun et al. 2021), which enables them to work in a normal room tem-

perature environment and in the presence of the Earth's magnetic field. This sets our MCG system apart from traditional MCG systems, which often require a cryogenic environment with liquid helium or a near-zero magnetic field environment with magnetic shielding. These improvements are necessary to make the practical MCG-based ID system.

However, since the MCG signals are much weaker than the environmental magnetic flux density, IFN and other high-frequency noises are significantly stronger than the MCG signals. There are three steps in denoising. Since the MCG signals of adjacent sampling points have physical relationships, the MCG signals of different sampling points are combined into three-dimensional matrices according to their positions as the data set for machine learning.

We use CNNs to classify individual MCG signals, making it a potential physiological feature for an ID system. The accuracy of individual identification is up to 97.04% by training the MCG data of five subjects under different environmental magnetic flux densities.

## Ethical Clearance

The collection and use of biological data in this paper was subject to ethical review under approval number IRB00001052-20120.

## Method and Algorithm

### MCG System

Our system (shown in Appendices Figure **??**) has two Bell-Bloom OPMs as a gradiometer to sense the cardiac magnetic field. It works at room temperature and in natural magnetic environments. The magnetometers are self-constructed and mounted in a 3D-printed structure with non-magnetic components such as mirrors, polarizers, wave plates, prisms, etc. A cesium vapor cell is coated with paraffin to increase the relaxation time of the coherent state of the cesium atoms, with a 7 Hz magnetic-resonance linewidth. A circularly-polarized pump laser is modulated on and off at the Larmor frequency with a duty cycle of 20%, and a continuous linearly-polarized probe laser is used to detect the atomic magnetic moment. The frequencies of both the pump and probe lasers are actively stabilized.

### Data Collection

We record the magnetic flux density perpendicular to the body plane (z-axis) of human subjects under environmental magnetic flux densities of 8,000 nT, 20,000 nT, 40,000 nT, 47,000 nT, 60,000 nT, and 80,000 nT. This simulates the influence of the angle between the sensitive direction of the OPMs and the direction of the geomagnetic field on the magnetic field data.

To capture the differences in MCG signals at different positions, we measure the MCG signals at 49 different locations above the thorax, as shown in Figure 1. The projections of the magnetic field vectors on the z-axis are significantly different at different positions (Maslennikov et al. 2012).

To optimize the quality of magnetocardiography (MCG) signals, we collected both electrocardiogram (ECG) and finger pulse signals simultaneously with the MCG signals. By

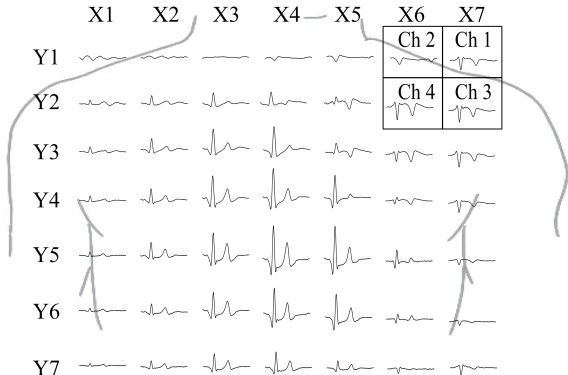

Figure 1: The coordinates of all the points for the measurement. From Y1 to Y7 is aligned along the spine with adjacent rows 5 cm apart and row Y1 aligned with the clavicle. From X1 to X7 is arranged perpendicular to the spine with a distance of 5 cm between adjacent columns. The midpoint of the interval between X3 and X4 is aligned with the intersection of the two clavicles. MCG signals at 49 positions are obtained based on the $7 \times 7$ measurement matrix. The denoised MCG diagrams are placed on the corresponding coordinates to show the cardiac magnetic field in front of the chest cavity. The box represents the position relationship between different data channels when MCG signals are converted into deep learning data sets.

comparing our MCG signal with other traditional cardiac signals, we confirmed that our MCG signal accurately reflects the true signal of the subject's heart.

Our approach of collecting multiple cardiac signals during MCG recordings is particularly effective in addressing the challenges associated with noise that can impact MCG signal quality. By comparing the MCG signal with the ECG and finger pulse signals, we were able to confirm the accuracy and reliability of our MCG signal, thus providing a strong foundation for future research and clinical applications.

To minimize the influence of factors such as gender, age, body type, disease, and others, we measure four healthy male subjects with similar body sizes, aged 23, 21, 20, and 17, as well as a 60-year-old healthy male subject. The data collected from these subjects are added to our dataset.

### Data Processing

**Denoising** The raw data collected from the MCG measurement has an amplitude of 3,000 pT. The main source of noise is the 50 Hz industrial frequency and its related frequencies, which we try to remove. To minimize the impact on the MCG signal, we apply a 75 Hz low-pass filter and reduce the residual noise to about 40 pT. To further reduce the impact of the industrial frequency noise, we add a sine wave with adjustable parameters to the filtered data. The parameters of the sine wave are optimized every 5 seconds to account for the unstable amplitude and frequency of the industrial frequency noise.

Appendices Figure 3 shows an MCG waveform measured

at the point (X4, Y4) in an environmental magnetic field of 47,000 nT as an example. The frequency is displayed in Hz, and the power spectral density is displayed in pT per square Hz.The MCG waveform, not only contains the intrinsic body activity signal generated by the electrical activity of the heart and other organs, but also the related noise and the influences of the environmental magnetic field. The denoised data is displayed in Figure 3 (a). The magnetic induction intensity is displayed in pT and the time elapsed since the start of the measurement is displayed in seconds. A positive value indicates the magnetic induction intensity is pointing upwards relative to the thoracic plane, and a negative value indicates it's pointing downwards. The peak of the R-wave is about 20 pT, and the valley of the S-wave is about -5 pT. The power spectral density of the denoised data is presented in Figure 3 (b). Although some high-frequency noise still exists due to the industrial frequency noise being larger than the intrinsic body activity signal, the intrinsic body activity signal becomes the dominant component.

**Time-frequency matrix** We apply the wavelet transform to convert the MCG signal waveform into time-frequency matrices, with the wavelet basis function Symlet6 (Daamouche et al. 2012). This method is appropriate for extracting frequency domain information without losing the time domain information.

In the machine learning process, it is common to classify biological information by training 2D images of the time-frequency signals with a network structure designed for image recognition (Kim, Kim, and Pan 2020). Nevertheless, this method may not be able to extract information about the correlation between the vertical components of the MCG signal. To overcome this issue, we arrange the time-frequency matrices generated by wavelet transform according to their relative positions, and treat each set of four adjacent time-frequency matrices as one channel of data, as shown in Figure 1. This approach makes it possible in practice to use a $2 \times 2$ sensor array placed anywhere in front of the chest cavity for individual identification.

We randomly select 2601 out of 3276 sets of experimental data as the training dataset, and the remaining 675 sets as the testing dataset. The CNN structure we applied in this paper is shown in Table 1.

## Result and Discussion

### Testing accuracy

The important criteria to measure the quality of classification models are accuracy, precision, recall, and F1-score.

Firstly, we train that network structure with a double-classification job. The testing result is shown in TABLE 2.

The F1-score is 99.31%. By adjusting the classification threshold, we determine the performance of the ID system under different usage requirements. In the double-classification system, when the sensitivity is reduced to 98.5%, the specificity can reach 99.8%. When the specificity is reduced to 97.9%, the sensitivity can reach 100.0%. At different classification thresholds, the values of 1-sensitivity and 1-specificity are shown in Appendices Figure 4.

| Layers | Output size | |
|---|---|---|
| Convolution | 224×224 | |
| SE Layer | 224×224 | |
| Dense Block (1) | 224×224 | 1×1 conv, 3×3 conv |
| Transition Layer (1) | 112×112 | 2×2 ave pool |
| Dense Block (2) | 112×112 | 1×1 conv, 3×3 conv |
| Transition Layer (2) | 56×56 | 2×2 ave pool |
| Dense Block (3) | 56×56 | 1×1 conv, 3×3 conv |
| SE Layer | 56×56 | |
| Linear Layer | | fully-connected |

Table 1: The CNN structure applied in this paper. The DenseBlock is beneficial for extracting insignificant features, and the SE-module can improve the accuracy when the connection between different channels is meaningful. This network structure has some advantages in this task compared to the classical network structures.

| Pr \\ Ac | A | B | Prcsn% |
|---|---|---|---|
| A | 149 | 0 | 100.0 |
| B | 2 | 140 | 98.6 |
| Rcll% | 98.7 | 100.0 | |

Table 2: The precision and recall of double-classification. The identification result of two subjects indicates that the individual information can be found in MCG signal. Ac: Actual, Pr: Predicted, Rcll: Recall, Prscn:Precision.

Then we apply the same method for a multi-classification. The testing result is shown in TABLE 3.

As a result, we compute the macro F1-score of multi-classification. The macro F1-score is 97.04%, which means it can classify individuals effectively.

### Robust

The important criteria to measure the quality of classification models are accuracy, precision, recall, and F1-score. Since the experimental data selected for this paper are obtained in a stable magnetic field environment, it is necessary to verify the robustness of the experimental results. We retest the performance of the module utilizing noisy data. The random noise and Gaussian noise are added to the MCG signals (Phukpattaranont 2014; Wright et al. 2008).

**Random Noise** To more accurately represent the magnetic field noise that can arise when the detector is in a non-stationary state, random noise is incorporated into the wavelet-transformed time-frequency matrices. By incorporating this type of noise, the signal processing algorithm can be evaluated under conditions that better represent the complex and dynamic nature of real-world scenarios. In order to examine the impact of different levels of noise on the accuracy of the algorithm, a range of random noise intensities spanning from 0 to 10 $dB^{-1}$ is introduced into the signal. This range of intensities has been selected to enable a sys-

Table 3: The precision and recall of multi-classification. The identification result of five subjects is not as good as the result of two subjects. Nevertheless, the result still indicate that the individual identification based on MCG is feasible. Ac: Actual, Pr: Predicted, Rcll: Recall, Prscn:Precision.

| | Actual A | Actual B | Actual C | Actual D | Actual E | Total Precision | |
|---|---|---|---|---|---|---|---|
| Precision A | 173 | 0 | 1 | 2 | 3 | 179 | 96.65% |
| Precision B | 1 | 138 | 0 | 3 | 1 | 143 | 96.50% |
| Precision C | 0 | 0 | 30 | 0 | 0 | 30 | 100.00% |
| Precision D | 0 | 1 | 0 | 163 | 1 | 165 | 98.79% |
| Precision E | 5 | 0 | 1 | 1 | 151 | 158 | 95.57% |
| Total Recall | 179 | 139 | 32 | 169 | 156 | 675 | |

tematic exploration of the algorithm's performance across a broad spectrum of noise levels. It is important to note that all of the random noise added to the signal possesses the same standard deviation, which ensures consistency across all levels of noise intensity.

**Gaussian Noise** In order to better simulate the natural noise present in the Earth's geomagnetic field, Gaussian noise is incorporated into the wavelet-transformed time-frequency matrices. This approach is employed in recognition of the fact that the natural noise present in such fields can often have Gaussian distributions. To enable a systematic and thorough investigation of the influence of different levels of noise on the performance of the signal processing algorithm, a range of Gaussian noise intensities spanning from 0 to 10 $dB^{-1}$ is added to the signal. It is worth noting that all the noise introduced into the signal possesses the same standard deviation, as this ensures the noise characteristics are consistent across all levels of noise intensity. This standardized approach helps to minimize the potential for confounding factors to impact the accuracy of the findings.

**Robustness of MCD-ID system** With the previously trained model, we test the dataset with noise. The relationship between the test accuracy and the signal-to-noise ratio (SNR) of the noise added is shown in Figure 5.

The results of the robustness test demonstrate that our proposed identification method is capable of successful operation when the level of noise is significantly weaker than the MCG signal. Specifically, our findings suggest that the identification method performs well in scenarios where the level of noise is less than 2 $dB^{-1}$ relative to the MCG signal. It is worth noting that in such scenarios, the classification method is able to function effectively and accurately identify the underlying signals. However, as the intensity of the noise increases beyond this threshold, the accuracy of the classification method is significantly compromised. At noise levels exceeding 2 $dB^{-1}$, the classification method is unable to accurately distinguish between the signal and the noise, leading to a decrease in performance.

## Discussion

The feasibility of using MCG signals to create an identification system has been demonstrated, using data collected from subjects lying down and measured with a fixed probe under a stable magnetic field. There are still some problems. The results of the test in healthy men indicate that the MCG

signals can be utilized for individual identification. However, since it has proved that the CNN architecture can identify heart disease (Attia et al. 2019), there is a potential issue with the training method used in this paper, as it could result in MCG information related to heart disease being recorded as individual identification information, potentially leading to others with the same heart disease being incorrectly identified. This could lead to others with the same heart disease being mistakenly identified as the subject due to their shared pathological characteristics. Therefore, the effects of heart disease on the ID system need to be studied. In addition, the MCG signals of subjects with pneumoconiosis may be significantly different from normal subjects(Freedman, Robinson, and Johnston 1980). There is concern that individuals with pneumoconiosis who live in extreme environments may not be accurately identified using this system.

We collect the MCG signal from multiple points to standardize our data collection. However, relying solely on the MCG ID system does not guarantee that the data is obtained from these points. To implement this system successfully, we need to test various data collection matrices, such as a triangular matrix or a more complex matrix. The robustness test proposed in this paper is only a preliminary assessment. To validate the performance of this model in real-world scenarios, we must test it against other types of noise as well.

## Conclusion

Our self-built MCG system has enabled us to achieve individual identification without the need for a magnetically shielded room, even when operated at room temperature, demonstrating its successful implementation. We have demonstrated that MCG signals captured on a grid matrix placed in front of the human chest can be used to identify individuals. We collected MCG data from five subjects, comprising 3,276 3-second-long recordings, which were converted into $240 \times 240$ time-frequency matrices using wavelet transforms. By adjusting the classification threshold to balance sensitivity and specificity, we achieved an F1-score of 97.04% on the training dataset. Testing the model on a separate dataset of 675 recordings yielded an accuracy rate of 97.03%. Moreover, we confirmed that this model has reliable classification performance even in the presence of noise levels similar to or less than the signal. Our work not only provides a potential new method for individual recognition, but also advances the application of MCG in everyday life.

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

# Appendices

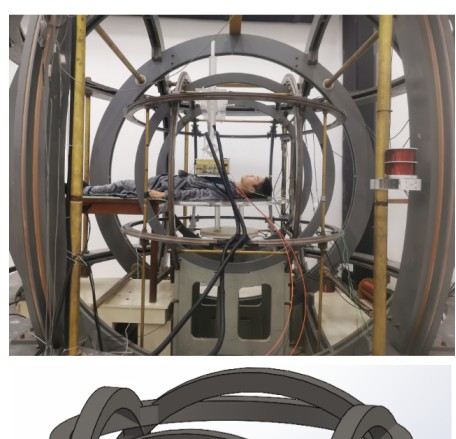

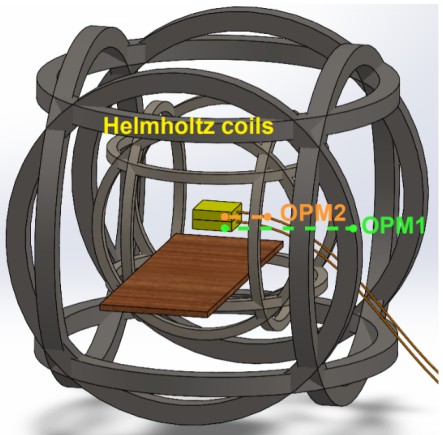

Figure 2: The photograph and schematic diagram of our MCG system. It consists of two Bell-Bloom OPMs as a gradiometer and a set of two-layer 3D Helmholtz coils. The gradiometer is applied to sense the cardiac magnetic field, at room temperature and in natural magnetic environments. The coils adjusts the direction and strength of the bias magnetic field and keeps the magnetic field around the vapor cell to a setpoint.

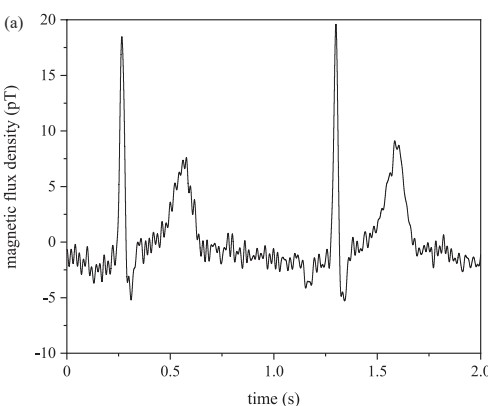

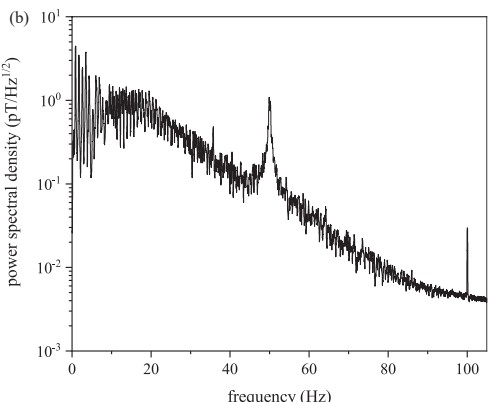

Figure 3: MCG signal was measured at the point (X4, Y4) in an environmental magnetic field of 47000 nT. Since the IFN is significantly larger than the intrinsic body activity signal, some high-frequency noise still exists in the MCG signal after filtering. (a) About 2 seconds of MCG signal after denoising. We present the magnetic induction intensity obtained by the probe in pT, and the time elapsed since the measurement began in second, respectively. A positive value and a negative value indicate that the direction of the magnetic induction intensity is perpendicular to the thoracic plane upward and downward, respectively. The peak of the R-wave is about 20 pT, and the peak of the S-wave is about -5 pT. (b) The power spectral density of MCG data after denoising, power spectral density is frequency. IFN still has an effect, but the intrinsic body activity signal is the major component. We present the frequency in Hz and the power spectral density corresponding to the frequency in pT over square Hz, respectively.

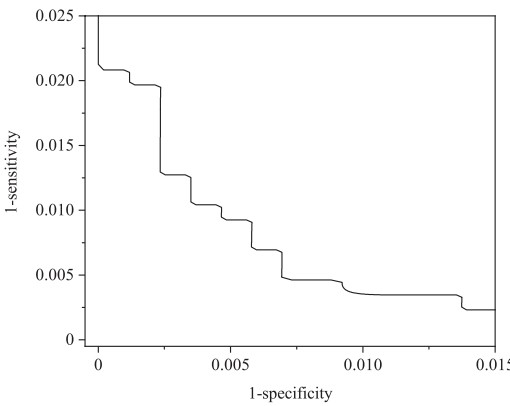

Figure 4: The sensitivity curve in double-classification. Changing the classification threshold resulted in a less difference in accuracy, indicating that most of the test data sets are significantly classified.

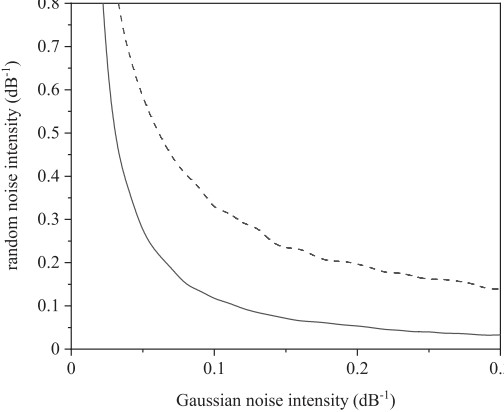

Figure 5: The performance of the trained module in classification when two types of noise are added simultaneously. We present the Gaussian noise intensity and the random noise intensity, respectively. The intensity of noise is defined as the bottom of the SNR. The real line and the dashed line represent the maximum acceptable noise threshold when the requirement for recognition accuracy is above 95% and 85%, respectively. The model is more robust to the Gaussian noise following a normal distribution than the random noise.