# OpenReview forum: "Identity information based on human magnetocardiography signals"
_AAAI.org/2024/Spring_Symposium_Series/Clinical_FMs — AAAI 2024 SSS on Clinical FMs_

### Official Review · Reviewer_bMgS · 2024-02-22
**Interesting topic but more experiments are needed**

**Rating:** 4
**Confidence:** 4

**Review:**

The work proposed an interesting work on using MCG signals to conduct individual identification. While several more steps needs to be considered as a foundation model

1.  "Our system (shown in Appendices Figure ??) has two BellBloom OPMs as a gradiometer to sense the cardiac magnetic
field." There is a reference issue.

2. More related work about MCG signal can be introduced to give readers a more comprehensive view of MCG.

3. In the introduction, the authors state that ECG quality can be influenced by physical activities etc, will MCG will also be influenced by those factors?

4. The most concerning issue is the model is tested on 5 subjects, it can hardly be called Foundation model.

---

### Official Review · Reviewer_Ag5k · 2024-02-23
**The research addresses a significant issue by exploring alternative methods for identity identification using MCG. Nowadays, security and privacy concerns are inevitable, and innovative approaches to identity verification are essential. By investigating the feasibility of extracting biometric information from magnetocardiography signals, the study offers an alternative for enhancing security measures without compromising individual privacy.**

**Rating:** 7
**Confidence:** 4

**Review:**

1.	The effort to control confounding factors by selecting participants with similar characteristics in this preliminary study is understandable. However, it's important to consider the potential trade-off between controlling variables and ensuring the model's generalizability to the broader population. With similar age and sex, the model may face increased difficulty in identifying individuals due to potential underlying differences in ECG patterns among different age groups and sexes. Given the limited diversity in the sample, there may be challenges in extrapolating the findings to a more diverse population. Future research could explore strategies to balance control over confounding factors with the need for dataset diversity to enhance the model's applicability across different demographics.

2.	Please further clarify the methodology employed to augment the dataset. Specifically, how were the repeated measures conducted and combined to generate such a large dataset (3276 sets) with 5 participants? Providing insights into this process would enhance the transparency of the study and offer valuable guidance to researchers interested in implementing similar data augmentation techniques in their work.

3.	I recommend the inclusion of measurements from ECG devices for comparison with the innovative self-manufactured device. To strengthen the evidence of the device's validity, it would be beneficial to authors include a figure comparing the measurements obtained from the new device with those from traditional ECG. This visual comparison could provide valuable insights into the device's accuracy and reliability.

4.	To ensure the robustness of the findings, it's crucial to provide details on the validation process for the model. Please elaborate on the validation dataset used to assess the model's performance.

5.	I suggest the authors consider using universal anatomical terms, such as transverse, coronal, and sagittal planes, to describe the human anatomy. Additionally, providing a fixed reference point when describing magnet induction intensity would be beneficial to prevent confusion and ensure the reproducibility and interpretation of the results. These adjustments could enhance clarity and consistency in the terminology used.

6.	I appreciate the authors' insightful approach to adding noise to mimic real-world environments. This strategy is particularly valuable as it reflects the common scenario where sensors encounter significant noise levels alongside the signal of interest. By incorporating noise into the study, the authors have taken a crucial step towards enhancing the realism and applicability of their findings to practical settings.

---

### Official Review · Reviewer_Uorw · 2024-02-23
**Interesting approach, relevance to topics in clinical foundation models is questionable**

**Rating:** 3
**Confidence:** 4

**Review:**

In this work, the authors develop and present preliminary results regarding an identification system based on magnetocardiography. The method for using MCG is an interesting non-invasive technique and significant effort has been shown in developing the system. However, there are several comments to be made regarding the work in the context of the symposium:

- First, the limited sample subjects does question the robustness of the results. Evaluation over a larger cohort would strengthen the conclusions of the paper
- The modality of MCG is novel, but given the size of the apparatus shown, it is questionable where such a system would be practical and needed.
- In addition, although the authors claim that MCG-based identification would be more robust and reliable, it is uncertain whether this is true given a lack of comparisons to other mentioned methods, such as facial features or ECG.
- The authors utilize a CNN with features extracted from a wavelet transform for the identification, it’s unclear that such a method would be scalable to large populations that would be necessary for practical deployment.
- Moreover, there doesn’t seem to be any significant innovation related to MCG or ML for healthcare in general.

Although the method of MCG is novel and interesting, further results and evaluation to other existing approaches is recommended